# Updated Risk Assessment of Cannabidiol in Foods Based on Benchmark Dose–Response Modeling

**DOI:** 10.3390/molecules29194733

**Published:** 2024-10-07

**Authors:** Eva Wisotzki, Heike Franke, Constanze Sproll, Stephan G. Walch, Dirk W. Lachenmeier

**Affiliations:** 1Postgraduate Study of Toxicology and Environmental Protection, Rudolf-Boehm-Institut für Pharmakologie und Toxikologie, Universität Leipzig, Härtelstrasse 16–18, 04107 Leipzig, Germany; eva.wisotzki@web.de (E.W.); heike.franke@medizin.uni-leipzig.de (H.F.); 2Chemisches und Veterinäruntersuchungsamt (CVUA) Karlsruhe, Weissenburger Strasse 3, 76187 Karlsruhe, Germany; constanze.sproll@cvuaka.bwl.de (C.S.); stephan.walch@cvuaka.bwl.de (S.G.W.)

**Keywords:** cannabidiol, novel food, health-based guidance value, benchmark dose, risk assessment

## Abstract

Cannabidiol (CBD), a non-psychotropic main component of the *Cannabis* plant, has been approved as a drug in the European Union (EU) under the name “Epidyolex”. However, its approval process as a food ingredient under the Novel Food Regulation was paused by the European Food Safety Authority (EFSA) due to a lack of safety data. Nevertheless, there is a growing, unregulated market in which CBD is advertised with various health claims and dosage instructions. Of particular concern is its toxic effect on the liver and possible reproductive toxicity in humans. Studies suitable for calculating the benchmark dose were identified from the available data. Animal studies yielded a benchmark dose lower confidence limit (BMDL) of 43 mg/kg bw/day, which translates into a safe human dose of approximately 15 mg/day. Only the Lowest-Observed-Adverse-Effect Level (LOAEL) of 4.3 mg/kg bw/day could be identified from the human data. This updated risk assessment confirmed a health-based guidance value (HBGV) of 10 mg/day based on human LOAEL. Despite the existing data gaps, preliminary regulation appears advisable because the current form of the gray CBD market is unacceptable from the standpoint of consumer safety and protection.

## 1. Introduction

Cannabidiol (CBD) (Figure 1) is a major component of the *Cannabis* plant (Figure 2) alongside tetrahydrocannabinol (THC). Unlike THC, CBD has no psychotropic effects, but its products are claimed to have various nutritional and health benefits.

Due to its strong lipophilic character, CBD has a low oral bioavailability [1]. When dissolved in oil or combined with a high-fat meal, bioavailability can be significantly increased [1,2]. In addition, chronic consumption can cause the substance to accumulate in fatty tissue, leading to long exposure windows [1,3].

As a multitarget drug, CBD interacts with various receptors throughout the body. In addition to the cannabinoid receptors CB1 and CB2, the compound also acts on receptors of various ligand-gated ion channels (e.g., glycine and GABA receptors such as GABA_A_) as well as agonistically on TRPV receptors such as TRPV1, on nuclear receptor proteins like PPARγ receptors, and it has a modulating effect on various G-protein-coupled receptors (e.g., adenosine and serotonin receptors, GPR55) [4]. Through its modulating effect on GABAergic neurotransmission, for example, it can affect the excitability of the central nervous system even in individuals with defective GABA_A_ receptors, making it an interesting candidate for the treatment of neurological disorders [5].

CBD is subject to high first-pass metabolism (Figure 3) by CYP450 enzymes (especially CYP2C19, CYP3A4) and UGT enzymes (UGT1A9, UGT2B7) [2]. Of the large number of metabolites, the main ones are 7-hydroxy cannabidiol (7-OH-CBD), 7-carboxy cannabidiol (7-COOH-CBD), 6-hydroxy cannabidiol (6-OH-CBD), and 4-hydroxy cannabidiol (4-OH-CBD), with 7-OH-CBD being the primary active metabolite [6,7] with a pharmacological activity similar to that of CBD [8]. The conversion of CBD to 7-OH-CBD is mainly catalyzed by CYP2C19 [6,7,8], although CYP2C9 may also play a role [6,8]. The ratio of the concentrations of CBD and 7-OH-CBD shifts toward CBD with prolonged use of CBD [2,9]. The active metabolite is further oxidized to the inactive 7-COOH-CBD by CYP2C19, CYP2D6 [6], and CYP3A4 [6,8]. CBD is also converted by CYP2C19 and CYP3A4 to the minor metabolites 6-OH-CBD and 4-OH-CBD [7]. Levels of 7-COOH-CBD increase after several weeks of CBD use and can still be detected in plasma 50 days after the last dose of CBD [9]. In women, the concentration of 7-COOH-CBD is higher than that in men, which may be due to the higher proportion of fat tissue in women [8,9]. There appears to be little difference between the different metabolic phenotypes [2,9]. This may be partly because CBD and its metabolites reversibly inhibit CYP enzymes. CBD can inhibit CYP2C19 and CYP3A isoforms, and 7-OH-CBD can inhibit CYP2C9 and CYP2D6 [10]. The effects of these metabolites are not yet fully understood and are still being studied.

Under the trade name “Epidyolex” in the European Union (EU) and “Epidiolex” in the United States (US), a prescription CBD medicinal product has been approved as an orphan drug for the treatment of rare epilepsy syndromes, namely Lennox–Gastaut syndrome and Dravet syndrome, in children aged two years and older. Subsequently, another indication for tuberous sclerosis was added [2].

In contrast to the well-defined legal framework for medicinal products, the regulatory landscape for CBD as a food supplement in the EU is markedly different. Since the consumption of CBD as a pure substance or in the form of CBD-rich hemp extracts has not been documented before 15 May 1997, the substance is categorized as novel in the European Commission’s Novel Food Catalogue [11] and requires pre-market approval under the Novel Food Regulation (Regulation (EU) 2015/2283) [12]. In the European Food Safety Authority (EFSA) statement from 2022, various data gaps were identified, which is why authorization has not yet been granted [13]. Therefore, foods and food supplements containing CBD are not legally marketable in the EU. In the US, CBD from industrial hemp with a maximum THC content of 0.3% was legalized at the federal level by the Agriculture Improvement Act of 2018 (also known as the Farm Bill). The US Food and Drug Administration (FDA), on the other hand, has only approved the drug Epidiolex. According to the FDA, the use of CBD in food and dietary supplements is illegal because it is a pharmacologically active ingredient in the approved drug Epidiolex, and as such is excluded from being a dietary supplement [14].

Of particular concern regarding foods containing CBD is liver toxicity at higher doses [15] and possible reprotoxic effects [16,17]. In contrast to pharmaceuticals, there is no compromise between efficacy and the potential for adverse effects in the context of foods. Foods must not be injurious to health [18].

Despite regulatory constraints, there has been a significant proliferation of CBD products in European and North American markets. They are most commonly offered in the form of oils, but also as capsules, gummy bears, or chocolates. The aim of this study was to update the risk assessment of CBD in light of new study results published since the authors’ last evaluation in 2023 [19].

## 2. Materials and Methods

### 2.1. Market and Consumer Analysis

To gain a comprehensive understanding of the current state of the CBD market with a principal focus on the European and American markets, a search was conducted on PubMed and Google Scholar using the term “cannabidiol” and keywords such as “market”, “advertising”, and “label accuracy”. To understand the motivation of users, surveys providing relevant information about the reasons for their consumption and use of cannabidiol were searched. Relevant articles were searched for citations that provided further information. To obtain an impression of the amount of CBD consumed, a search for “CBD dosage” was performed on Google on 2 December 2023, and various websites for recommended dosages were reviewed.

### 2.2. Literature Search Strategy for Risk Assessment

Because this study focused on CBD as a food, this research was limited to orally administered CBD alone. To focus on the effects of CBD, studies in which CBD was taken together with THC or other co-medications were excluded.

To gain insight into the toxicity of CBD, a search was conducted on PubMed and Google Scholar for the terms “cannabidiol safety” and “cannabidiol toxicity”. Meta-reviews and the most recent findings were given particular attention, with a focus on liver toxicity and reproductive toxicity. Further risk assessment was carried out based on the EFSA guidelines.

### 2.3. Health-Based Guidance Value (HBGV) Calculation

In order to support a regulatory evaluation of CBD as a novel food, a health-based guidance value (HBGV) for CBD was calculated based on the results of the most recent studies that were not covered by the last study that had already calculated an HBGV [19]. To calculate the HBGV, it is necessary to identify the benchmark dose (BMD) or No-Observed-Adverse-Effect Level (NOAEL) as a suitable reference point (RP). If this is not possible, the Lowest-Observed-Adverse-Effect Level (LOAEL) may be used as an alternative. The EFSA recommends, if possible, to use the BMD approach instead of the NOAEL approach as RP for risk assessments [20]. For the BMD approach, the guidance of the US Environmental Protection Agency (EPA) was followed [21,22].

#### 2.3.1. Identification of Human Data for HBGV Assessment

To identify suitable human studies, “Clinical Trial” and “Randomized Controlled Trial” as the article type were selected in PubMed (as of 22 May 2024), and a search for the term “cannabidiol safety” was conducted. A total of 121 results were obtained, and studies that did not meet the specified criteria were excluded. Seven studies were excluded because CBD was only used topically. In 43 of the discarded studies, other concomitant cannabinoids, mainly THC, were tested in addition to CBD. Another six studies included concomitant medications. Fifty-one studies were not eligible because there were no adequate dose groups and often only one or two different doses were tested. In addition, four further studies lacked a comparison with placebo and could not be used. One study was duplicated, and one study each was eliminated for the following reasons: it was only a study design, a study on a cell-based model, or had nothing to do with CBD. Two studies were eliminated because they were reviews.

Four studies met all predetermined qualifications and were retained for final analysis [23,24,25,26]. Three of the studies [23,24,25] were single-dose studies without dose by weight (mg/day instead of mg/kg bw/day). However, no serious adverse effects were observed in these studies. The final study [26] was conducted on children aged 4–10 years old, and adverse effects were not clearly related to the dose–response relationship. It is unfortunate that a BMD cannot yet be calculated using human studies. Therefore, the NOAEL approach must therefore be used. The NOAEL or LOAEL for liver toxicity in humans was derived from a meta-study and the HBGV was calculated using the uncertainty factors (UFs) recommended by EFSA [27].

#### 2.3.2. Identification of Animal Data for HBGV Assessment and Approach for BMD Modeling

For BMD to be calculated from data from animal experiments, the following criteria must be met [21,22]: the observed effects must be stated as mean values with associated deviation, the measured biological effect must be in a dose–response relationship to the substance used, and the number of animals tested per group should also be stated. In addition to the control group, at least three different dose groups, expressed in mg/kg bw/day, should have been tested without concomitant medication or THC. In order to calculate a BMD for liver toxicity in animals, a search of PubMed and Google Scholar was conducted for animal studies that met the aforementioned criteria.

The benchmark dose approach was based on the EPA Benchmark Dose online tool and its Technical Guidance Manual [22]. The default program settings with a confidence level of 0.95 were used. The online tool calculated various statistical models and indicated whether they were viable. The most appropriate model was selected based on the lowest Akaike information criterion (AIC). This is a method for comparing different models. The lowest AIC value means that the amount of information lost in this model is lower than in the other models; therefore, it was considered the most informative model.

### 2.4. Manuscript Preparation and Language Editing

The initial draft of this manuscript was written by the authors. Subsequently, to enhance language quality and readability, we utilized Trinka (https://www.trinka.ai/), an AI-powered English language editing tool developed by Crimson AI Pvt. Ltd. (Mumbai, India). Trinka was applied in ‘Power Mode’ on 19 September 2024. The tool scanned 9228 words and suggested 458 revisions across various categories including articles, punctuation, prepositions, and word choice. All suggestions were manually reviewed and selectively implemented by the authors to ensure scientific accuracy and preserve the original meaning. The authors retain full responsibility for the content, including parts refined using this AI tool.

## 3. Results

### 3.1. Market and Consumers

Cannabidiol is becoming increasingly popular, and its market is growing. Sales in Europe are predicted to reach EUR 3.47 billion by 2024 [28]. Various surveys from Germany [29], the UK and Ireland [30], the USA [31,32], and Canada [32] have examined the number of users, reasons for use, and public perception of CBD. CBD appears to be most popular in the USA, where 26% of respondents [32] stated that they had used CBD, compared to 11% in Germany [29]. However, there is also some time between the studies (2022 USA, 2020 Germany), and the number of consumers in Germany may have increased in the meantime. The most prevalent method of oral CBD administration is via oils, tinctures, and edibles [29,31,33], and the Internet, including social media, appears to be a popular source of information [31,34,35].

One study published in 2021 searched the microblogging platform Twitter (now https://x.com) for information on CBD and found many health-related statements [36]. If cannabidiol is marketed as a food in the EU, the substance is subject to the Health Claims Regulation (EC) No. 1924/2006, which prohibits such claims for food and food supplements.

Another study examined the use of CBD on Pinterest [37]. CBD is mostly presented positively here, health-related statements are also made, and almost none of the so-called pins contain information on any adverse effects. Many of them linked to CBD online stores, followed by personal blog posts and other social media. Only approximately 1% link to medical websites or pharmaceutical companies. This discrepancy between the high volume of information generated by CBD retailers and the lack of evidence-based information on this social media platform was also reflected in the perception and risk assessment of the population. Over 50% of respondents rated CBD as a low-risk substance [29] that is “good” or “very good” for health [29,32], and many respondents also saw no problem in taking CBD together with prescription drugs [31]. In addition, efficacy claims are frequently based on personal experience reports [31,37] rather than medical evidence.

Most consumers use CBD to relieve pain [29,30,31,32,35,38] and to improve sleep [29,31,32,33,38] or mental health. The latter refers on the one hand to a stress-reducing [29,33,38] and relaxing effect of CBD [29,33], while on the other hand, it also refers to serious psychological problems such as depression [31,32,35] and anxiety [30,31,32,38]. Sometimes the prescribed medication is reduced or omitted in favor of CBD [31,38].

In addition to the illegality of health claims, the unregulated market presents another challenge. A significant number of studies have demonstrated that the CBD concentration in a product frequently does not align with the information displayed on the label [39,40,41,42]. In the best-case scenario, overlabeling may mislead consumers; in the worst case, underlabeling can result in overdosing and the development of adverse effects. A previous study [39] found that underlabeling occurred mainly with high-dose CBD oils. Beverages were more likely to be affected by underlabeling; in some cases, only traces of the substance below the limit of quantification were detected, possibly due to the poor water solubility of the substance. One study in Germany [43] examined 22 CBD oils and 4 oils of other phytocannabinoids. The actual concentration of cannabinoids was, on average, 21% higher than stated on the labeling. Because many people obtain information from the Internet, dosing information recommended on the web is summarized in Table 1.

Of the 26 first search hits, 24 were online stores for CBD, one was a newspaper website, and one was a job exchange website for doctors. Here, a wide range of dosages is of note, which are often divided into low, medium, and high doses. In addition, prohibited health claims can be found for specific doses. There is also frequent advice to increase the dosage gradually, starting with a low dose until the desired effect is achieved. However, there is currently no evidence of the efficacy of CBD for many indications. Consequently, in the case of diseases for which CBD has insufficient efficacy, consumers might continue to increase their doses in the hope of achieving a positive effect at some point.

The studies by Kaufmann et al. provide insight into the dosages actually used by long-term consumers, as all participants had already consumed CBD for at least 30 days (most often more than 12 months) before the start of the study and were able to determine their own dosage and product type freely. The average dosage in the studies was 50 mg with a standard deviation (SD) of approximately 40 mg [38,44,45], and with a range of 8–390 mg/day [38,44] and 2.5–390 mg/day [22], respectively.

### 3.2. Health Risks

The adverse effects of the prescription drug Epidiolex (dose range 10–25 mg/kg bw/day) include gastrointestinal discomfort, infections, loss of appetite, drowsiness and lethargy, seizure, cough, elevated liver enzyme levels, and rash [2]. Some in vitro and in vivo studies have shown a positive influence of CBD on various liver diseases, as it probably protects the liver from oxidative stress and has an effect on lipid metabolism and cell apoptosis of the liver [46]. In contrast, some clinical studies on humans showed that CBD might adversely increase liver enzymes. This is of particular concern when consuming CBD as a food supplement, because changes in liver enzyme levels, unlike the other adverse effects, may not be noticed by the consumer for a long time unless blood is regularly tested by a doctor.

The liver toxicity of CBD was investigated in a meta-analysis by Lo et al. [15]. CBD can increase liver values up to drug-induced liver injury (DILI), whereby the CBD-associated DILI is comparable to other liver-toxic compounds, such as statins and fluoroquinolones, and it occurs in both children and adults, including healthy adults. DILI was not studied at doses below 300 mg/day. The association between increased liver enzyme levels and DILI is particularly significant at CBD doses of >1000 mg/day or ≥20 mg/kg bw/day and concomitant use of valproic acid, a drug from the anticonvulsant group, which is also used for treating epilepsy. In most cases, the increase in liver enzyme levels is reversible after discontinuation of CBD; sometimes, the values even normalize spontaneously despite maintenance of the therapy [15].

Two studies by Kaufmann et al. examined the impact of long-term oral CBD consumption on liver health. The studies excluded people with known liver disease, liver dysfunction, and CBD allergy, as well as people taking certain medications, supplements, or foods known to affect the liver. A total of 1475 people were enrolled, of which 839 [45] or 1061 [44] people completed the studies. As mentioned in Section 3.1, the study participants were able to select the dose and product type themselves, and the daily dose ranged from 2.5 to 390 mg [44,45]. It is noticeable that the smallest amounts were taken by people who used nano-formulations, whereas the largest were taken by people who chose CBD as an additive or as a capsule/pill dosage form. In both studies, almost 10% of the participants showed an increase in the liver enzyme aspartate transaminase (ALT) level in the final blood test and approximately 4% showed an increase in the liver enzyme alanine transaminase (AST) level. Elevated ALT levels >3 × upper limit of normal (ULN) occurred in 3 [45] and 4 [44] individuals, respectively, and were associated with underlying diseases. The authors state that there was no relationship between the selected dose and ALT elevation, rather than demographic, physical, and medical conditions. These results were compared with the incidence of elevated liver enzyme levels in the general population as measured by the National Health and Nutrition Examination Survey (NHANES) from 1999 to 2002, and because of similar values, Kaufmann et al. claimed that CBD was not harmful to the liver at these doses. However, given the absence of comparison groups and the unavailability of blood values prior to long-term CBD consumption, the authors believe that it is not currently possible to make definitive statements regarding the impact of CBD on the test subjects’ livers. Comparisons with the general population are not feasible due to the exclusion criteria and the time difference of at least 10 years. Furthermore, it is not entirely clear why 222 additional subjects were included in one study. Caputi [47] also criticized the study design: On the one hand, observational studies may be inadequate to address the issue at hand; on the other hand, the majority of the study participants were women, who are less prone to liver problems. Furthermore, only subjects who had already consumed CBD over a longer period of time were admitted to the study; thus, individuals who did not tolerate the substance well were excluded from the study.

It is assumed that cannabidiol has toxic effects on reproduction in animals, but comprehensive studies on humans are lacking. A review by Carvalho et al. [16] investigates the effect of CBD on the reproductive system of male animals both at the genetic level and through enzyme induction and competition with other substrates for certain enzymes. Receptors and enzymes of the endocannabinoid system are believed to play a role in the hypothalamic–pituitary–gonadal axis and thus influence the production of sex hormones. Studies both in vivo and in vitro have shown that CBD can have a stimulating or inhibitory effect on CYP enzymes (such as the inhibition of CYP3A4, which is responsible for the metabolism of testosterone [6]) that control the breakdown of sex steroids. A biphasic effect of CBD has been shown in rodents: at low doses, it indirectly increases the concentration of endocannabinoids, which have an inhibitory effect on copulation, whereas at high doses, it can have a stimulatory effect on sexual behavior by blocking endocannabinoid enzymes and activating TRPV1 receptors. The substance presumably influences testosterone and androgen concentrations, as well as has a gonadotoxic effect on spermatogenesis. CBD consumption by the mother shortly before birth can induce long-term changes in the reproductive endocrine functions of male rodents. Unfortunately, due to the study design of the animal experiments, no statement on the reversibility of dysregulation of the reproductive system is possible.

Sarrafpour et al. [48] investigated the use of CBD in pregnant women. The use of CBD for morning sickness is popular. The high lipophilicity of CBD, which renders the substance potentially placental, could be problematic. The cannabinoid system plays a role in all stages of pregnancy, from conception to birth, and its effect on newborns and pregnant woman cannot be ruled out. CBD also has an anti-angiogenic effect on the endothelial cells of the human umbilical vein and can therefore cause pregnancy complications. CBD’s ability to modulate the immune system by influencing cytokine levels and apoptosis could also be a cause for concern. This could interfere with the development of the fetus’ immune system. Studies in an ex vivo model [49,50] have also shown that CBD can alter the function of two efflux transporters, namely the breast cancer resistance protein (BCRP) and the P-glycoprotein (P-gp), which can be inhibited by CBD and thus make the placental barrier more permeable to xenobiotics, ultimately harming the fetus.

### 3.3. Health-Based-Guidance-Value (HBGV)

The results of benchmark dose modeling are summarized in Table 2. The full BMD modeling results are provided in Appendix A. A BMDL of 43 mg/kg per day was the most appropriate and was therefore chosen as the reference point (RP). In order to calculate an HBGV using the BMDL of the animal studies as RP, it is necessary to apply an interspecies default factor of 10 and a human intraspecies default factor of 10 in order to extrapolate the effects of animal studies on rats to humans and extrapolate the differences between each human. Furthermore, a default factor of 2 is used to extrapolate from sub-chronic to chronic consumption [51].

An HBGV of 0.21 mg/kg bw/day can be calculated from the animal study [52] using the BMDL. This corresponds to a low-risk daily intake of approximately 15 mg. However, since this value is not based on human studies, it is more conservative to use human studies to define an appropriate reference point for the HBGV, even if this precludes the use of a BMDL for the time being.

An LOAEL of 300 mg/day was obtained from the meta-study by Lo et al. [15]. This is the lowest dose tested in humans at which DILI occurs. This value was derived from a subacute 28-day study by Crippa et al. [55] involving 59 healthy adults in which liver enzyme levels increased in four individuals at this dose (150 mg twice a day). The EFSA also confirmed this value as LOAEL [13]. Consequently, when extrapolating from an LOAEL to a NOAEL, an additional UF must be included [27], which must be decided on a case-by-case basis because there is no standard value. Therefore, the UF of 3 used by EFSA for THC, a structurally similar compound, was applied [56].

The EFSA is unable to provide default values for extrapolation from subacute studies to chronic intake. It should be noted that human studies conducted over longer periods do not indicate the specific time point at which the liver values of the test subjects were elevated. In most cases, however, the increase was detected in the fourth week [15]. Given these uncertainties and evidence that these liver problems may be reversible by cessation of CBD use, it is proposed that the extra UF may be excluded at this point.

The 300 mg dose can be converted to the body weight of a 70 kg individual, resulting in an approximate dose of 4.29 mg/kg bw/day. This resulted in an HBGV of 0.14 mg/kg bw/day (equivalent to approximately 10 mg per day) for temporary use of 2 weeks (Table 3).

Previously, Lachenmeier et al. [19] calculated a BMDL of 20 mg/kg bw/day based on a rat study, GWTX1412 [57], published by the FDA as part of the Epidiolex approval. This BMDL is judged to be of higher quality because the study included more animals per group, used purified CBD as the test compound, and was conducted over a longer period of 26 weeks. The endpoint was centrilobular hypertrophy of the liver. The HBGV calculated by Lachenmeier et al. is close to the candidate HBGV calculated in this study at 0.20 mg/kg/day or 14 mg/day.

The results of the available studies consistently indicate that a dose of approximately 10 mg/day may be the most appropriate level for HBGV. This value was derived from human studies and represents the lowest dose, which may exclude DILI risks for consumers. Taking CBD in foods above the HBGV is not recommended and may be associated with increased risks.

## 4. Discussion

Human studies are generally preferable to animal studies in health risk assessment. Unfortunately, the use of an LOAEL entails considerable uncertainties. However, animal studies appear to support this value, as evidenced by the results of this study and previous studies [19]. Based on a typical average daily consumption of 50 ± 40 mg [45], most consumers exceed the HBGV of 10 mg. Considering a recent survey of CBD products on the German market, an average of 2–7 drops of CBD oil per day would exceed the HBGV [43]. In addition, doses above the LOAEL are occasionally suggested on the Internet. It would therefore be advisable for competent authorities to examine these findings and consider taking measures to improve the current situation and minimize potential risks to the population.

As mentioned above, Lachenmeier et al. [19] calculated an HBGV for CBD in food. They also derived the same HBGV from Crippa et al. [55]. Their study also included the GWTX1412 [57] animal study mentioned above. However, no extra UF was used to extrapolate the study duration from sub-chronic to chronic. If a UF of 2 was included, a more conservative limit of 0.10 mg/kg/day or 7 mg/day would be obtained. However, this value is also close to the HBGV calculated in this study.

Similarly, Henderson et al. [58] calculated an acceptable daily intake (ADI) for the general population using human studies. Also, based on the study of Crippa et al. [55], a NOAEL of 300 mg/day was proposed as the RP or, in this case, point of departure (POD). As the authors are in agreement with EFSA that this value is an LOAEL and not a NOAEL, the UF for extrapolation from LOAEL to NOAEL is consequently missing in the calculation of Henderson et al. [58]. They only use a UF of 3 for inter-human variability. The rationale behind the deviation from the standard default value of 10 and the omission of pharmacodynamic differences between individuals remains unclear. Pharmacodynamics may vary due to differences in receptor density, hence the importance of gradual dose titration during therapy [1,59]. Pharmacokinetics may vary depending on genetics, weight, time of administration (fasting or with food), vehicle, and other factors [1]. Henderson et al. [58] have also proposed a recommended upper intake level (UL) for the consumption of CBD only by healthy adults who do not want to have children and are not pregnant or breastfeeding. The selected NOAEL chosen was 1000 mg/day, as reported by Lo et al. [15]. As this dosage is strongly associated with an increase in liver enzyme values, it is not suitable for the risk assessment of CBD in food, and the authors believe that the resulting UL of 1.42 mg/kg/day is not safe.

The Swiss Federal Office for Food Safety and Veterinary Medicine (BLV) issued a statement recommending that a daily oral dose of 12 mg should not be exceeded because of the liver-damaging effects of CBD [60]. This value was derived from a human study published in a document from GW Pharmaceuticals, the pharmaceutical company that manufactures Epidiolex. This 2019 document [61] states that Epidiolex’s approval studies showed DILI at 5 mg/kg/day. The BLV calculated the maximum daily dose of 12 mg using a UF of 10 for inter-human variability and 3 to reflect that effects occurred at the lowest dose tested. This is consistent with the HBGV of this study.

In the United Kingdom, the Committee on Toxicity (COT) of the Food Standards Agency (FSA) has initially set an upper daily intake of 70 mg CBD [62]. A CBD dose of ≤1 mg/kg was chosen for this value, because inhibitory interactions with some medications occurred at this dose. They then calculated this for a person with a standard weight of 70 kg. Based on new data, the COT more recently adjusted this value and set a provisional ADI of 0.15 mg/kg bw/day or 10 mg/day for a 70 kg person for pure CBD [63]. This value was calculated on the basis of three unpublished 90-day rodent studies submitted by the applicants for novel food authorization, which are unfortunately not publicly available. The NOAEL (72, 50, and 25 mg/kg bw/day) was taken as the POD based on the adverse liver effects observed in these three unpublished studies. Supported by the results of human studies, an average ADI of 10 mg/day was calculated from the ADIs (17, 12, and 5.6 mg per person) of the studies.

All recommended maximum daily doses are summarized in Table 4.

Although there are not many human studies on the low-dose CBD range, existing human studies [15,61] seem sufficient to assess the risk of liver effects, even at low doses, especially since human data are supported by animal studies. However, more research is needed to assess the potential harm to fertility. The endocannabinoid system plays an important role in all stages of pregnancy, from conception to birth, so its effects on newborns and pregnant women cannot be ruled out.

Another challenge is bioavailability, which can vary considerably depending on the fasting state and vehicle. One study demonstrated that the selection of vehicle oil for CBD could influence its bioavailability, not to mention CBD as a nanoformulation [64]. Many studies are currently underway in this area to increase bioavailability. The introduction of new formulations into the market may necessitate corresponding adjustments to the HBGV.

Another significant source of uncertainty is the presence of other cannabinoids in full-spectrum hemp extracts, which are frequently found alongside CBD. These compounds have been the subject of even less research than CBD, making it currently impossible to assess their exact effects and adverse effects.

While food supplements and medicinal products may contain CBD, their regulatory frameworks differ significantly. For medicinal products, certain adverse effects of CBD may be tolerated if the therapeutic benefits outweigh the risks. However, this risk–benefit approach is not applicable to food supplements, which are regulated as food. Consequently, CBD-containing food supplements require distinct risk assessments and must adhere to different regulatory standards than their medicinal counterparts.

However, the implementation of effective regulatory measures for CBD food products poses significant challenges. Although CBD products not authorized as novel foods are technically illegal under EU law, enforcement remains complex. In principle, national food safety authorities have sufficient means at their disposal to take action against unauthorized food supplements and unsafe foods. In Europe, for CBD supplements, these primarily include the EU Novel Food Regulation (Regulation (EU) 2015/2283) and the General Food Law Regulation (Regulation (EC) No 178/2002), particularly when products are considered unsafe. These regularly frameworks enable authorities to conduct market surveillance, issue product recalls, and impose penalties on non-compliant companies. The EU-wide Rapid Alert System for Food and Feed (RASFF) facilitates information sharing on non-compliant products across member states. However, the work of the authorities is often complicated by several factors. The popularity of CBD products, fueled by excessive advertising of supposed health benefits while risks remain unmentioned, poses a significant challenge. Additionally, regulating online marketplaces and cross-border sales presents unique difficulties. The rapid evolution of the CBD market can also outpace regulatory responses, further complicating enforcement efforts. Manufacturers and retailers bear a particular responsibility in this context. They are legally obligated to ensure only safe food products enter the market, a duty that becomes even more critical with novel ingredients like CBD. To address these challenges effectively, further legal analysis by regulatory experts is recommended. This could help develop more comprehensive and agile enforcement strategies that can keep pace with the dynamic CBD supplement market.

## 5. Conclusions

After a thorough evaluation of the latest study results on liver toxicity, animal and human studies seem to be consistent in their results, so an HBGV of 10 mg/day for CBD in food seems justified. However, this value is a preliminary recommendation based on the currently available data on CBD-induced liver damage. It would be a prudent precautionary public health protection measure to consider this value as a possible reference point for authorities, although further scientific research is required, especially in the low-dose range and in reproductive toxicology, as harmful effects on reproduction cannot currently be ruled out.

## Figures and Tables

**Figure 1 molecules-29-04733-f001:**
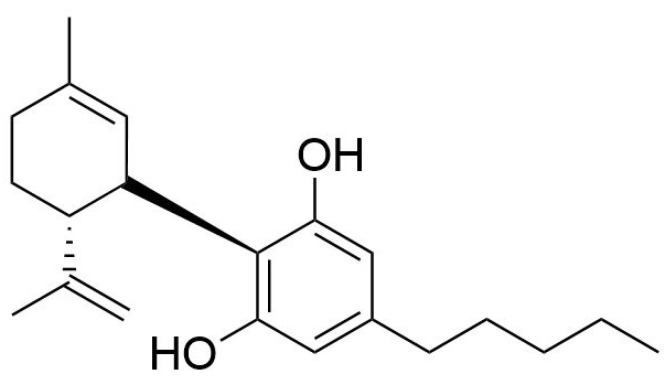
Structural formula of cannabidiol.

**Figure 2 molecules-29-04733-f002:**
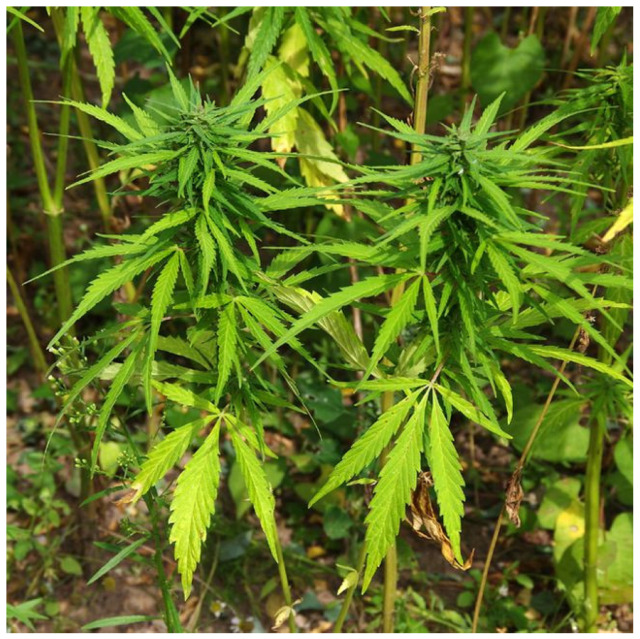
*Cannabis sativa* plant (own photography by H.F.).

**Figure 3 molecules-29-04733-f003:**
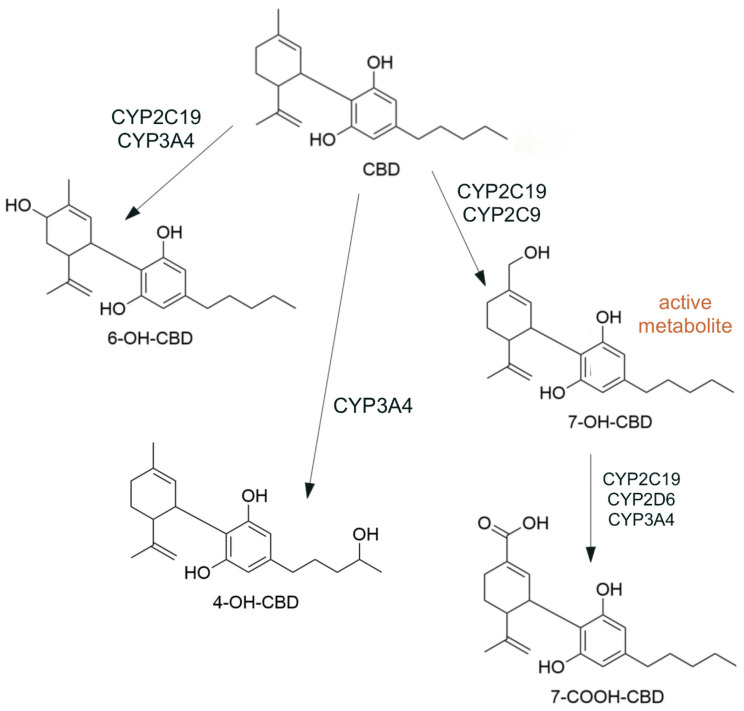
CBD and its main metabolites (phase 1 metabolism in the liver).

**Table 1 molecules-29-04733-t001:** CBD online dosing recommendations (full data in Appendix Table A1).

Dosage ^1^	Number of Investigated Websites (*n* = 26)	Mean Dosage (and Range) [mg/day]
Low dosage (microdosage)	13	15 (0.5–25)
Medium dosage (standard dosage)	13	47 (10–100)
High dosage (macrodosage)	12	227 (50–800)
Other dosage information	10	62 (2.5–300)
Medicinal indication statements ^2^	8	-

^1^ Dosage recommendations online often refer to low, medium, and high dose ranges, and titration to effective dose is often recommended. ^2^ Dosages are often mentioned with supposedly suitable indications for various diseases.

**Table 2 molecules-29-04733-t002:** Benchmark dose modeling of cannabidiol in animal experiments.

Study, Tested Substance	Species, Sex ^1^	Study Design, CBD Doses ^2^	Endpoint	BMDL	BMD	*p*-Value	Model ^3^
Henderson et al. [52] CBD-isolate	Rats, female	90-day oral toxicity study 0, 50, 80, 120, 140 mg/kg bw/day	liver weight	43 mg/kg bw/day	64 mg/kg bw/day	0.242	Power
Tallon and Child [53] Hemp extract (6.27% CBD)	Rats, female	90-day oral toxicity study 0, 30, 115.13, 230.25, 460.5 mg/kg bw/day	liver weight	(80 mg/kg bw/day) ^3^	(94 mg/kg bw/day) ^3^	0.948	Linear
Dziwenka et al. [54] Elixinol Hemp Extract (around 65% CBD)	Rats, male	90-day oral toxicity study 0, 18.95, 33.16, 56.84 mg/kg bw/day	liver weight	(30 mg/kg bw/day) ^4^	(41 mg/kg bw/day) ^4^	0.644	Exponential 3

^1^ The more sensitive sex was selected. ^2^ Dosages were adjusted to the pure CBD content specified in the study, if necessary. ^3^ Recommended model results (lowest Akaike Information Criterion (AIC)) were selected with BMDS online web-based application. ^4^ Not a pure CBD test substance; therefore, a general statement is not possible.

**Table 3 molecules-29-04733-t003:** Candidates for health-based guidance values (HBGVs) from animal and human studies.

	Human Study	Animal Study
Study	Crippa et al. [55]	Henderson et al. [52]
Reference point (RP)	LOAEL of 300 mg/day (=4.29 mg/kg bw/day)	BMDL of 43 mg/kg bw/day
Uncertainty factor (UF)	30 ^1^	200 ^2^
HBGV	0.14 mg/kg bw/day (10 mg/day)	0.21 mg/kg bw/day (15 mg/day)

^1^ 10 for intraspecies variability × 3 for extrapolation from LOAEL to NOAEL. ^2^ 10 for interspecies variability × 10 for intraspecies variability × 2 for extrapolation from sub-chronic to chronic.

**Table 4 molecules-29-04733-t004:** Summary of health-based guidance values (HBGVs) for oral cannabidiol intake.

Study	Health-Based Guidance Value/ Acceptable Daily Intake	Rationale
COT FSA [62] (2021)	70 mg/day	1 mg/kg bw/day for inhibitory interactions in humans
BLV [60] (2021)	12 mg/day	5 mg/kg bw/day for DILI in humans
Lachenmeier et al. [19] (2023)	10 mg/day	300 mg/day for DILI in humans
Henderson et al. [58] (2023)	30 mg/day	300 mg/day for DILI in humans
COT FSA [63] (2023)	10 mg/day	72, 50, and 25 mg/kg bw/day for adverse liver effects in rodents
This study	10 mg/day	300 mg/day for DILI in humans

## Data Availability

Publicly available datasets were analyzed in this article. New derivative data presented in this article are available in the Appendix A.

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
