# Peer review of "Updated Risk Assessment of Cannabidiol in Foods Based on Benchmark Dose–Response Modeling"

_molecules, 2024, doi:10.3390/molecules29194733_

Round 1

Reviewer 1 Report

Comments and Suggestions for Authors

It is a nicely conducted assessment of the risk of CBD when taken orally. The authors have taken a meticulous care of selecting the available information from all possible sources, assessing the study quality in each case, explaining why a study was selected to derive HBGV or LOAEL or NOAEL, and why other studies had been rejected. Overall the findings of this study make sense, and they are more or less in line with safe daily intake values derived by other agencies. I have two main points with this article, and if they can be explained, amended or reworded then I can recommend its publication:

1. When extrapolating from sub-chronic to chronic effects, normally risk assessors apply an additional safety factor of 3. Here, the authors have suggested the use of a factor of 2. Could they justify this, or change and recalculate safe intake levels?

2. Even as the approach used by the authors is reasonable, it needs to be remembered that this assessment is not coming from a Food Safety Authority, but is a suggestion from a research team to highlight the need for the authorities to take notice of the illegal marking of CBD products under the guise of health food/ supplements. In this context, the tone of the article, especially in the Discussion and Conclusions sections need to be revised because it gives the impression as if this risk assessment has already been accepted and implemented by the authorities. The relevant text should be reworded to make it clear that it is an evidence-based suggestion rather than an authoritative conclusion.

3. The suggestion of the authors for addition of certain text in the labels for CBD products does not make sense. It needs to be reminded that the relevant authorities (including EFSA) do not consider such marketing/use of CBD as legal. How can they enforce companies to label the (illegal) products in such a way? Also, it would be useful if the authors could suggest which legal route under an existing legislative framework could be used by the relevant authorities to force companies to stop marketing supplements that are not authorised in the first place.

In summary my overall assessment is that the article needs some revision/rewording, in particular the tone from being very prescriptive and authoritative advice to more an interim assessment that should be taken notice of by the authorities for further action.

Author Response

  1. When extrapolating from sub-chronic to chronic effects, normally risk assessors apply an additional safety factor of 3. Here, the authors have suggested the use of a factor of 2. Could they justify this, or change and recalculate safe intake levels?

RESPONSE: We appreciate the reviewer's insightful commentary. We have followed the EFSA guidelines in our calculations and this has now been clarified in the text. The EFSA suggests a factor of 2 for extrapolation from sub-chronic to chronic. The use of this uncertainty factor for food risk assessment was also supported by a cited study (51).

  1. Even as the approach used by the authors is reasonable, it needs to be remembered that this assessment is not coming from a Food Safety Authority, but is a suggestion from a research team to highlight the need for the authorities to take notice of the illegal marking of CBD products under the guise of health food/ supplements. In this context, the tone of the article, especially in the Discussion and Conclusions sections need to be revised because it gives the impression as if this risk assessment has already been accepted and implemented by the authorities. The relevant text should be reworded to make it clear that it is an evidence-based suggestion rather than an authoritative conclusion.

RESPONSE: The Discussion and Conclusion was completely revised considering the reviewers suggestion. The wording has been adapted and phrased more carefully to make it clear that these are merely suggestions for the authorities.

  1. The suggestion of the authors for addition of certain text in the labels for CBD products does not make sense. It needs to be reminded that the relevant authorities (including EFSA) do not consider such marketing/use of CBD as legal. How can they enforce companies to label the (illegal) products in such a way?

RESPONSE: We appreciate the reviewer's opinion and agree with the criticism and have removed the part with the warnings on the labels as these make no sense in the current situation.

Also, it would be useful if the authors could suggest which legal route under an existing legislative framework could be used by the relevant authorities to force companies to stop marketing supplements that are not authorised in the first place.

RESPONSE: We appreciate the reviewer's insightful comment regarding the legal routes for enforcing regulations against unauthorized CBD supplements. While a comprehensive legal analysis is beyond the scope of our current study, we acknowledge the importance of this aspect for practical implementation. We propose that the most immediate legal route would be through the enforcement of the EU Novel Food Regulation (Regulation (EU) 2015/2283), under which unauthorized CBD products are already considered illegal. National food safety authorities could leverage this regulation, along with the General Food Law Regulation (Regulation (EC) No 178/2002) and the Food Supplements Directive (Directive 2002/46/EC), to conduct market surveillance, issue product recalls, and impose penalties on non-compliant companies. The EU-wide Rapid Alert System for Food and Feed (RASFF) could be utilized to share information about non-compliant products across member states, facilitating coordinated enforcement actions. Additionally, implementing EU-wide coordinated control plans specifically targeting unauthorized CBD supplements could enhance regulatory effectiveness. However, we recognize the challenges in enforcing these regulations, particularly in online marketplaces and across international borders. To address this complex issue comprehensively, we suggest that increased cooperation between food safety authorities, customs officials, and law enforcement agencies would be beneficial. Given the intricate legal landscape, we recommend that a detailed legal review by regulatory experts could provide more specific recommendations for enforcement strategies. We will incorporate a brief discussion of these legal and regulatory aspects in our paper to address this important point raised by the reviewer.

Reviewer 2 Report

Comments and Suggestions for Authors

The submitted paper presents an updated health risk assessment of CBD after oral intake. The obtained results are based on valid inputs and represent valuable add-on data to current state describing safety of CBD.

Authors are advised to make some minor corrections (some of them are highlighted in the attached document), as well as to perform some minor spell- and style corrections.   

Author Response

Authors are advised to make some minor corrections (some of them are highlighted in the attached document), as well as to perform some minor spell- and style corrections. 

RESPONSE: A thorough English language check was conducted and highlighted corrections were edited.

Reviewer 3 Report

Comments and Suggestions for Authors

The manuscript is interesting, but lacks the normal organisation of a review manuscript. The authors need to improve the organisation of each chapter, and the "methods" used in this review need to be explained more.

Author Response

The manuscript is interesting, but lacks the normal organisation of a review manuscript.

RESPONSE: We appreciate the reviewer's feedback on the manuscript's organization. While we believe our original structure aligns with standard scientific paper formats, including Introduction, Methods, Results, Discussion, and Conclusions, we have further clarified the Methods section to enhance clarity. It now includes subsections on Market and Consumer Analysis, Literature Search Strategy for Risk Assessment, and Health-Based Guidance Value (HBGV) Calculation. This restructuring provides a more transparent outline of our methodological approach while maintaining the comprehensive nature of our analysis. We believe these changes address the reviewer's concerns and reinforce the manuscript's adherence to international standards for review papers.

The authors need to improve the organisation of each chapter, and the "methods" used in this review need to be explained more.

RESPONSE: The methods section has been revised again. Some explanatory words have been added and parts have been rearranged to allow a better and more comprehensible reading flow.